# Unsupervised Hyperalignment for Multilingual Word Embeddings

## Abstract

We consider the problem of aligning continuous word representations, learned in multiple languages, to a common space. It was recently shown that, in the case of two languages, it is possible to learn such a mapping without supervision. This paper extends this line of work to the problem of aligning multiple languages to a common space. A solution is to independently map all languages to a pivot language. Unfortunately, this degrades the quality of indirect word translation. We thus propose a novel formulation that ensures composable mappings, leading to better alignments. We evaluate our method by jointly aligning word vectors in eleven languages, showing consistent improvement with indirect mappings while maintaining competitive performance on direct word translation.

## 1 Introduction

Pre-trained continuous representations of words are standard building blocks of many natural language processing and machine learning systems (Mikolov et al., 2013b). Word vectors are designed to summarize and quantify semantic nuances through a few hundred coordinates. Such representations are typically used in downstream tasks to improve generalization when the amount of data is scarce (Collobert et al., 2011). The distributional information used to learn these word vectors derives from statistical properties of word co-occurrence found in large corpora (Deerwester et al., 1990). Such corpora are, by design, monolingual (Mikolov et al., 2013b; Bojanowski et al., 2016), resulting in the independent learning of word embeddings for each language.

A limitation of these monolingual embeddings is that it is impossible to compare words across languages. It is thus natural to try to combine all these word representations into a common multilingual space, where every language could be mapped. Mikolov et al. (2013a) observed that word vectors learned on different languages share a similar structure. More precisely, two sets of pre-trained vectors in different languages can be aligned to some extent: a linear mapping between the two sets of embeddings is enough to produce decent word translations. Recently, there has been an increasing interest in mapping these pre-trained vectors in a common space (Xing et al., 2015b; Artetxe et al., 2017), resulting in many publicly available embeddings in many languages mapped into a single common vector space (Smith et al., 2017; Conneau et al., 2017; Joulin et al., 2018). The quality of these multilingual embeddings can be tested by composing mappings between languages and looking at the resulting translations. As an example, learning a direct mapping between Italian and Portuguese leads to a word translation accuracy of $78.1\%$ with a nearest neighbor (NN) criterion, while composing the mapping between Italian and English and Portuguese and English leads to a word translation accuracy of $70.7\%$ only. Practically speaking, it is not surprising to see such a degradation since these bilingual alignments are trained separately, without enforcing transitivity.

In this paper, we propose a novel approach to align multiple languages simultaneously in a common space in a way that enforces transitive translations. Our method relies on constraining word translations to be coherent between languages when mapped to the common space. Nakashole and Flauger (2017) has recently shown that similar constraints over a well chosen triplet of languages improve supervised bilingual alignment. Our work extends their conclusions to the unsupervised case. We show that our approach achieves competitive performance while enforcing composition.

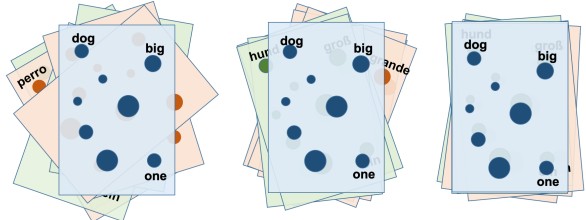

Figure 1: Left: we do not impose constraint for language pairs not involving English, giving poor alignments for languages different than English. We can add constraints between languages belonging to a same family (middle) or between all pairs (right), leading to better alignments.

## 2 PRELIMINARIES ON BILINGUAL ALIGNMENT

In this section, we provide a brief overview of bilingual alignment methods to learn a mapping between two sets of embeddings, and discuss their limits when used in multilingual settings.

### 2.1 SUPERVISED BILINGUAL ALIGNMENT

Mikolov et al. (2013a) formulate the problem of word vector alignment as a quadratic problem. Given two sets of $n$ word vectors stacked in two $n \times d$ matrices $\mathbf{X}$ and $\mathbf{Y}$ and a $n \times n$ assignment matrix $\mathbf{P} \in \{0,1\}^{n \times n}$ built using a bilingual lexicon, the mapping matrix $\mathbf{Q} \in \mathbb{R}^{d \times d}$ is the solution of the least-square problem: $\min_{\mathbf{Q} \in \mathbb{R}^{d \times d}} \|\mathbf{XQ} - \mathbf{PY}\|_2^2$, which admits a closed form solution. Restraining $\mathbf{Q}$ to the set of orthogonal matrices $\mathcal{O}_d$, improves the alignments (Xing et al., 2015b). The resulting problem, known as Orthogonal Procrustes, still admits a closed form solution through a singular value decomposition (Schnemann, 1966).

**Alternative loss function.** The $\ell_2$ loss is intrinsically associated with the nearest neighbor (NN) criterion. This criterion suffers from the existence of "hubs", which are data points that are nearest neighbors to many other data points (Dinu et al., 2014). Alternative criterions have been suggested, such as the inverted softmax (Smith et al., 2017) and CSLS (Conneau et al., 2017). Recently, Joulin et al. (2018) has shown that directly minimizing a loss inspired by the CSLS criterion significantly improve the quality of the retrieved word translations. Their loss function, called RCSLS, is defined as:

$$\text{RCSLS}(\mathbf{x}, \mathbf{y}) = -2\mathbf{x}^\top \mathbf{y} + \frac{1}{k} \sum_{\mathbf{y} \in \mathcal{N}_Y(\mathbf{x})} \mathbf{x}^\top \mathbf{y} + \frac{1}{k} \sum_{\mathbf{x} \in \mathcal{N}_X(\mathbf{y})} \mathbf{x}^\top \mathbf{y}. \tag{1}$$

This loss is a tight convex relaxation of the CSLS cristerion for normalized word vectors, and can be efficiently minimized with a subgradient method.

### 2.2 UNSUPERVISED BILINGUAL ALIGNMENT: WASSERSTEIN-PROCRUSTES

In the setting of unsupervised bilingual alignment, the assignment matrix $\mathbf{P}$ is unknown and must be learned jointly with the mapping $\mathbf{Q}$. An assignment matrix represents a one-to-one correspondence between the two sets of words, i.e., is a bi-stochastic matrix with binary entries. The set of assignment matrices $\mathcal{P}_n$ is thus defined as:

$$\mathcal{P}_n = \mathcal{B}_n \cap \{0,1\}^{n \times n}, \text{ where } \mathcal{B}_n = \left\{ \mathbf{P} \in \mathbb{R}_+^{n \times n}, \mathbf{P1}_n = \mathbf{1}_n, \ \mathbf{P}^\top \mathbf{1}_n = \mathbf{1}_n \right\}.$$

The resulting approach, called Wasserstein-Procrustes (Zhang et al., 2017a; Grave et al., 2018), jointly learns both matrices by solving the following problem:

$$\min_{\mathbf{Q} \in \mathcal{O}_d} \min_{\mathbf{P} \in \mathcal{P}_n} \|\mathbf{XQ} - \mathbf{PY}\|_2^2. \tag{2}$$

This problem is not convex since neither of the sets $\mathcal{P}_n$ and $\mathcal{O}_d$ are convex. Minimizing over each variable separately leads, however, to well understood optimization problems: when $\mathbf{P}$ is fixed, minimizing over $\mathbf{Q}$ involves solving the orthogonal Procrustes problem. When $\mathbf{Q}$ is fixed, an optimal permutation matrix $\mathbf{P}$ can be obtained with the Hungarian algorithm. A simple heuristic to address Eq.(2) is thus to use an alternate optimization. Both algorithms have a cubic complexity but on different quantities: Procrustes involves the dimension of the vectors, i.e., $O(d^3)$ (with $d = 300$), whereas the Hungarian algorithm involves the size of the sets, i.e., $O(n^3)$ (with $n = 20k$–$200k$).

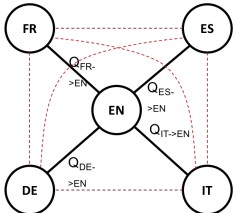

Figure 2: Plain black lines indicate pairs of languages for which a translation matrix is learned. Dashed red lines indicate pairs used in the loss functions. Languages are aligned onto a common pivot language, i.e., English.

Directly applying the Hungarian algorithm is computationally prohibitive, but efficient alternatives exist. Cuturi (2013) shows that regularizing this problem with a negative entropy leads to a Sinkhorn algorithm, with a complexity of $O(n^2)$ up to logarithmic factors (Altschuler et al., 2017).

As for many non-convex problems, a good initial guess helps converge to better local minima. Grave et al. (2018) compute an initial $\mathbf{P}$ with a convex relaxation of the quadratic assignment problem. We found, however, that the entropic regularization of the Gromov-Wasserstein (GW) problem (Mémoli, 2011) worked well in practice and was significantly faster (Solomon et al., 2016; Peyré et al., 2016):

$$\min_{\mathbf{P} \in \mathcal{P}_n} \sum_{i,j,i',j'} (\|\mathbf{x}_i - \mathbf{x}_{i'}\|_2 - \|\mathbf{y}_j - \mathbf{y}_{j'}\|_2)^2 P(i,j) P(i',j') + \epsilon \sum_{i,j} P(i,j) \log P(i,j).$$

The case $\varepsilon = 0$ corresponds to Mémoli's initial proposal. Optimizing the regularized version ($\epsilon > 0$) leads to a local minimum $\hat{\mathbf{P}}$ that can be used as an initialization to solve Eq. (2). Note that a similar formulation was recently used in the same context by Alvarez-Melis and Jaakkola (2018).

## 3 Composable Multilingual Alignments

In this section, we propose an unsupervised approach to jointly align $N$ sets of vectors to a unique common space while preserving the quality of word translation between every pair of languages.

### 3.1 Multilingual alignment to a common space

Given $N$ sets of word vectors, we are interested in aligning these to a common target space $\mathcal{T}$. For simplicity, we assume that this target space coincide with one of the word vector set. The language associated with this vector set is denoted as "pivot" and is indexed by $i = 0$. A typical choice for the pivot, used in publicly available aligned vectors, is English (Smith et al., 2017; Conneau et al., 2017; Joulin et al., 2018). Aligning multiple languages to a common space consists in learning a mapping $\mathbf{Q}_i$ for each language $i$, such that its vectors are aligned with the vectors of the pivot language up to a permutation matrix $\mathbf{P}_i$:

$$\min_{\mathbf{Q}_i \in \mathcal{O}_d, \ \mathbf{P}_i \in \mathcal{P}_n} \quad \sum_i \ell(\mathbf{X}_i \mathbf{Q}_i, \mathbf{P}_i \mathbf{X}_0), \tag{3}$$

This objective function decomposes over each language and does not guarantee good indirect word translation between pairs of languages that do not include the pivot. A solution would be to directly enforce compositionality by adding constraints on the mappings. However, this would require to introduce mappings between all pairs of languages, leading to the estimation of $O(N^2)$ mappings simultaneously. Instead we leverage the fact that all the vector sets are mapped to a common space to enforce good alignments within this space. With the convention that $\mathbf{Q}_0$ is equal to the identity, this leads to the following problem:

$$\min_{\mathbf{Q}_i \in \mathcal{O}_d, \ \mathbf{P}_{ij} \in \mathcal{P}_n} \quad \sum_{i,j} \alpha_{ij} \ell(\mathbf{X}_i \mathbf{Q}_i, \mathbf{P}_{ij} \mathbf{X}_j \mathbf{Q}_j), \tag{4}$$

where $\alpha_{ij} > 0$ weights the importance of the alignment quality between the languages $i$ and $j$. This formulation does not introduce any unnecessary mapping. It constrains all pairs of vector sets to be well aligned, instead of directly constraining the mappings. Constraining the mappings would encourage coherence over the entire space, while we are interested in well aligned data; that is coherent mapping within the span of the word vectors. Our approach takes its inspiration from the hyperalignment of multiple graphs (Goodall, 1991). We refer to our approach as Unsupervised Multilingual Hyperalignment (UMH).

**Choice of weights.** The weights on the alignments can be chosen to reflect prior knowledge about the relations between languages. For example, these weights can be a function of a rough language similarity measure extracted from the initial alignment. It is not clear however how the weights should depend on these similarities: constraining similar languages with higher weights may be unnecessary, since they are already well aligned. On the other hand, increasing weights of distant languages may lead to a problem where alignments are harder to learn. In practice, we find that simple weighting schemes work best with little assumptions. For instance, uniform weights work well but tends to degrade the performance of translations to and from the pivot if the number of languages is large. Instead, we choose to use larger weights for direct alignments to the pivot, to insure that our multilingual objective does not degrade bilingual alignments to the pivot. In practice, $\alpha_{ij}$ is set to $N$ if $i$ or $j$ is equal to 0, and 1 otherwise.

**Choice of loss function.** We consider the RCSLS loss of Joulin et al. (2018). This loss is computationally expensive and wasteful to minimize it from scratch. We thus optimize a $\ell_2$ for the first couple of epochs before switching to the RCSLS loss. This two-step procedure shares some similarities with the use of a refinement step after learning a first rough alignment (Artetxe et al., 2017; Conneau et al., 2017). The $\ell_2$ loss between two sets of normalized vectors is equivalent to a linear function, up to an additive constant $C$:

$$\ell_2(\mathbf{QX}, \mathbf{PY}) = -2\mathrm{tr}\left(\mathbf{Q}^T\mathbf{X}^T\mathbf{PY}\right) + C \tag{5}$$

Where $\mathbf{Q} = \mathbf{Q}_i\mathbf{Q}_j^T$ if $\mathbf{X}$ and $\mathbf{Y}$ are the vectors from languages $i$ and $j$. We adapt the RCSLS loss of Eq. (1) to the unsupervised case by applying an assignment matrix to the target vectors:

$$\mathrm{RCSLS}(\mathbf{XQ}, \mathbf{PY}) = -2\mathrm{tr}\left(\mathbf{Q}^\top\mathbf{X}^\top\mathbf{PY}\right) + \frac{1}{k}\left[\sum_{\substack{\mathbf{x}\in\mathbf{X}, \\ \mathbf{z}\in\mathcal{N}_{\mathbf{PY}}(\mathbf{Q}^\top\mathbf{x})}} \mathbf{z}^\top\mathbf{Q}^\top\mathbf{x} + \sum_{\substack{\mathbf{y}\in\mathbf{PY}, \\ \mathbf{Q}^\top\mathbf{z}\in\mathcal{N}_X(\mathbf{y})}} \mathbf{z}^\top\mathbf{Qy}\right].$$

**Efficient optimization.** Directly optimizing Eq. (4) is computationally prohibitive since $N^2$ terms are involved. We use a stochastic procedure where $N$ pairs $(i, j)$ of languages are sampled at each iteration and updated. We sample pairs according to the weights $\alpha_{ij}$. The RCSLS loss being slower to compute than the $\ell_2$ loss, we first optimize the latter for a couple of epochs before switching to the former. The $\ell_2$ loss is optimized with the same procedure as in Grave et al. (2018): alternate minimization with a Sinkhorn algorithm for the assignment matrix on small batches. We then switch to a cruder optimization scheme when minimizing the RCSLS loss: we use a greedy assignment algorithm by taking the maximum per row. We also subsample the number of elements to compute the $K$ nearest neighbors from. We restrict each set of vectors to its first 20k elements. UMH runs on a CPU with 10 threads in less than 10 minutes for a pair of languages and in 2 hours for 6 languages.

## 4 RELATED WORK

**Bilingual word embedding alignment.** Since the work of Mikolov et al. (2013a), many have proposed different approaches to align word vectors with different degrees of supervision, from fully supervised (Dinu et al., 2014; Xing et al., 2015a; Artetxe et al., 2016; Joulin et al., 2018) to little supervision (Smith et al., 2017; Artetxe et al., 2017) and even fully unsupervised (Zhang et al., 2017a; Conneau et al., 2017; Hoshen and Wolf, 2018). Among unsupervised approaches, some have explicitly formulated this problem as a distribution matching: Cao et al. (2016) align the first two moments of the word vector distributions, assuming Gaussian distributions. Others (Zhang et al., 2017b; Conneau et al., 2017) have used a Generative Adversarial Network framework (Goodfellow et al., 2014). Zhang et al. (2017a) shows that an earth mover distance can be used to refine the alignment obtained from a generative adversarial network, drawing a connection between word embedding alignment and Optimal Transport (OT). Artetxe et al. (2018) proposes a stable algorithm to tackle distant pairs of languages and low quality embeddings. Closer to our work, Grave et al. (2018) and Alvarez-Melis and Jaakkola (2018) have proposed robust unsupervised bilingual alignment methods based on OT. Our approach takes inspiration from their work and extend them to the multilingual setting.

**Multilingual word embedding alignment.**  Nakashole and Flauger (2017) showed that constraining coherent word alignments between triplets of nearby languages improves the quality of induced bilingual lexicons. Jawanpuria et al. (2018) recently showed similar results on any triplets of languages in the supervised case. As opposed to our work, these approaches are restricted to triplets of languages and use supervision for both the lexicon and the choice of the pivot language. Finally, independently of this work, Chen and Cardie (2018) has recently extended the bilingual method of Conneau et al. (2017) to the multilingual setting.

**Optimal Transport.**  Optimal transport (Villani, 2003; Santambrogio, 2015) provides a natural topology on shapes and discrete probability measures (Peyré et al., 2017), that can be leveraged with fast OT problem solvers (Cuturi, 2013; Altschuler et al., 2017). Of particular interest is the Gromov-Wasserstein distance (Gromov, 2007; Mémoli, 2011). It has been used for shape matching under its primitive form (Bronstein et al., 2006; Mémoli, 2007) and under its entropy-regularized form (Solomon et al., 2016). We use the latter for our intialization.

**Hyperalignment.**  Hyperalignment, as introduced by Goodall (1991), is the method of aligning several shapes onto each other with supervision. Recently, Lorbert and Ramadge (2012) extended this supervised approach to non-Euclidean distances. We recommend Gower et al. (2004) for a thorough survey of the different extensions of Procrustes and to Edelman et al. (1998) for algorithms involving orthogonal constraints. For unsupervised alignment of multiple shapes, Huang et al. (2007) use a pointwise entropy based method and apply it to face alignment.

## 5 EXPERIMENTAL RESULTS

**Implementation Details.**  We use normalized fastText word vectors trained on the Wikipedia Corpus (Bojanowski et al., 2016). We use stochastic gradient descent (SGD) to minimize the RCSLS loss. We run the first epoch with a batch size of $500$ and then set it to 1k. We set the learning rate to $0.1$ for the $\ell_2$ loss and to $25$ for the RCSLS loss in the multilingual setting, and to $50$ in the bilingual setting. For the first two iterations, we learn the assignment with a regularized Sinkhorn. Then, for efficiency, we switch to a greedy assignment, by picking the max per row of the score matrix. We initialize with the Gromov-Wasserstein approach applied to the first 2k vectors and a regularization parameter $\varepsilon$ of $0.5$ (Peyré et al., 2016). We use the python optimal transport package. [1]

**Extended MUSE Benchmark.**  We evaluate on the MUSE test datasets (Conneau et al., 2017), learning the alignments on the following 11 languages: Czech, Danish, Dutch, English, French, German, Italian, Polish, Portuguese, Russian and Spanish. MUSE bilingual lexicon are mostly translations to or from English. For missing pairs of languages (e.g., Danish-German), we use the intersection of their translation to English to build a test set. MUSE bilingual lexicon are built with an automatic translation system. The construction of the new bilingual lexicon is equivalent to translate with pivot.

**Baselines.**  We consider as baselines several bilingual alignment methods that are either supervised, i.e., Orthogonal Procrustes, GeoMM (Jawanpuria et al., 2018) and RCSLS (Joulin et al., 2018), or unsupervised, i.e., Adversarial (Conneau et al., 2017), ICP (Hoshen and Wolf, 2018), Gromov-Wasserstein (GW) (Alvarez-Melis and Jaakkola, 2018) and Wasserstein Procrustes ("Wass Proc.") (Grave et al., 2018). We also compare with the unsupervised multilingual method of Chen and Cardie (2018). All the unsupervised approaches, but GW, apply the refinement step (ref.) of Conneau et al. (2017) or of Chen and Cardie (2018).

### 5.1 TRIPLET ALIGNMENT

In these experiments, we evaluate the quality of our formulation in the simple case of language triplets. One language acts as the pivot between the two others. We evaluate both the direct translation to and from the pivot and the indirect translation between the two other languages. An indirect translation is obtained by first mapping the source language to the pivot, and then from the pivot to

---

[1] POT, https://pot.readthedocs.io/en/stable/

|  | Direct | | Ind. | Direct | | Ind. | Direct | | Ind. |
| --- | --- | --- | --- | --- | --- | --- | --- | --- | --- |
|  | de-en | fr-en | de-fr | de-fr | pt-es | fr-es | pt-fr | pt-fr | fi-en | hu-en | fi-hu | fi-hu |
| Pairs | **72.3** | **80.2** | 64.5 | 61.7 | 86.5 | **81.2** | 77.2 | 72.3 | **53.4** | **55.9** | 45.6 | 31.9 |
| Triplet | 71.9 | **80.2** | - | **68.3** | **86.8** | **81.2** | - | **77.9** | 50.2 | **55.9** | - | **42.4** |

Table 1: Accuracy averaged on both directions (source→target and target→source) with a NN criterion on triplet alignment with direct translation ("Direct") and indirect translation ("Ind."). Indirect translation uses a pivot (source→pivot→target). The pivot language is underlined. We compare our approach applied to pairs ("Pairs") of languages and triplets ("Triplets").

|  | en-es | | en-fr | | en-it | | en-de | | en-ru | | Avg. |
| --- | --- | --- | --- | --- | --- | --- | --- | --- | --- | --- | --- |
|  | → | ← | → | ← | → | ← | → | ← | → | ← | |
| *supervised, bilingual* | | | | | | | | | | | |
| Proc. | 80.9 | 82.9 | 81.0 | 82.3 | 75.3 | 77.7 | 74.3 | 72.4 | 51.2 | 64.5 | 74.3 |
| GeoMM | 81.4 | 85.5 | 82.1 | 84.1 | - | - | 74.7 | 76.7 | 51.3 | 67.6 | - |
| RCSLS | 84.1 | 86.3 | 83.3 | 84.1 | 79.3 | 81.5 | 79.1 | 76.3 | 57.9 | 67.2 | 77.9 |
| *unsupervised, bilingual* | | | | | | | | | | | |
| GW | 81.7 | 80.4 | 81.3 | 78.9 | 78.9 | 75.2 | 71.9 | 72.8 | 45.1 | 43.7 | 71.0 |
| Adv. + ref. | 81.7 | 83.3 | 82.3 | 82.1 | 77.4 | 76.1 | 74.0 | 72.2 | 44.0 | 59.1 | 73.2 |
| ICP + ref. | 82.1 | 84.1 | 82.3 | 82.9 | 77.9 | 77.5 | 74.7 | 73.0 | **47.5** | 61.8 | 74.4 |
| W-Proc. + ref. | 82.8 | 84.1 | 82.6 | 82.9 | - | - | **75.4** | 73.3 | 43.7 | 59.1 | - |
| UMH bil. | **82.5** | 84.9 | **82.9** | **83.3** | 79.4 | 79.4 | 74.8 | 73.7 | 45.3 | 62.8 | 74.9 |
| *unsupervised, multilingual* | | | | | | | | | | | |
| MAT+MPSR | **82.5** | 83.7 | 82.4 | 81.8 | 78.8 | 77.4 | 74.8 | 72.9 | - | - | - |
| UMH multi. | **82.4** | **85.1** | 82.7 | **83.4** | 78.1 | **79.3** | **75.5** | **74.4** | 45.8 | **64.9** | **75.2** |

Table 2: Accuracy of supervised and unsupervised approaches on the MUSE benchmark. All the approaches use a CSLS criterion. "ref." refers to the refinement method of Conneau et al. (2017). UMH does not used a refinement step. Multilingual ("Multi.") UMH is trained on 11 languages simultaneously. The best overall accuracy is underlined, and in bold among unsupervised methods.

the target language. For completeness, we report direct translation between the source and target languages. This experiment is inspired by the setting of Nakashole and Flauger (2017).

Table 1 compares our approach trained on language pairs and triplets. We use a NN criterion to give insights on the quality of the dot product between mapped vectors. We test different settings: we change the pivot or the pair of languages, consider natural in-between and distant pivot. We also consider languages harder to align to English, such as Finnish or Hungarian.

Overall, these variations have little impact on the performance. The direct translation to and from the pivot is not significantly impacted by the presence or absence of a third language. More interestingly, the indirect translation of a model trained with 3 languages often compares favorably with the direct translation from the source to the target. In comparison, the performance of indirect translation obtained with a bilingual model dropped by $6 - 8\%$. This drop reduces to a couple of percents if a CSLS criterion is used instead of a NN criterion.

## 5.2 MULTILINGUAL ALIGNMENT

In this set of experiments, we evaluate the quality of joint multilingual alignment on a larger set of 11 languages. We look at the impact on direct and indirect alignments.

**Direct word translation.** Table 2 shows a comparison of UMH with other unsupervised approaches on the MUSE benchmark. This benchmark consists of 5 translations to and from English. In this experiment, UMH is jointly trained on 10 languages, plus English. The results on the remaining 5 languages are in the appendix. We observe a slight improvement of performance

|  | Latin | Germanic | Slavic | Latin-Germ. | Latin-Slavic | Germ.-Slavic | All |
|---|---|---|---|---|---|---|---|
| *supervised, bilingual* | | | | | | | |
| Proc. | 75.3 | 51.6 | 47.9 | 50.2 | 46.7 | 40.9 | 50.7 |
| *unsupervised, bilingual* | | | | | | | |
| W-Proc.* | 74.5 | 53.2 | 44.7 | 52.2 | 44.6 | 40.2 | 50.3 |
| UMH Bil. | 76.6 | 54.6 | 45.5 | 53.7 | 46.3 | 40.8 | 51.7 |
| *unsupervised, multilingual* | | | | | | | |
| UMH Multi. | **79.0** | **58.8** | **49.8** | **57.8** | **48.8** | **45.4** | **55.3** |

Table 3: Accuracy with a NN criterion on indirect translations averaged among and across language families. The languages are Czech, Danish, Dutch, English, French, German, Italian, Polish, Portuguese, Russian and Spanish. UMH is either applied independently for each pairs formed with English ("Bil.") or jointly ("Multi."). W-Proc.* is our implementation of Grave et al. (2018) with a Gomorov-Wasserstein initialization and our optimization scheme.

of $0.3\%$ compared to bilingual UMH, which is also consistent on the remaining 5 languages. This improvement is not significant but shows that our approach maintains good direct word translation.

**Indirect word translation.** Table 3 shows the performance on indirect word translation with English as a pivot language. We consider averaged accuracies among and across language families, i.e. Latin, Germanic and Slavic. As expected, constraining the alignments significantly improves over the bilingual baseline, by almost $4\%$. The biggest improvement comes from Slavic languages. The smallest improvement is between Latin languages ($+2\%$), since they are all already well aligned with English. In general, we observe that our approach helps the most for distant languages, but the relative improvements are similar across all languages.

## 5.3 ABLATION STUDY

In this section, we evaluate the impact of some of our design choices on the performance of UMH. We focus in particular on the loss function, the weighting and the initialization. We also discuss the impact of the number of languages used for training on the performance of UMH.

**Impact of the loss function.** Table 2 compares bilingual UMH, with state-of-the-art unsupervised bilingual approaches on the MUSE benchmark. All the approaches use the CSLS criterion. UMH directly learns a bilingual mapping with an approximation of the retrieval criterion. Bilingual UMH compares favorably with previous approaches ($+0.5\%$). In particular, the comparison with "W-Proc.+ref" validates our choice of the RCSLS loss for UMH.

|  | Direct | Indirect |
|---|---|---|
| Uniform | 65.5 | **56.9** |
| UMH | **69.4** | 55.3 |

Table 4: Comparison of uniform weights and our weighting in UMH on indirect and direct translation with a NN criterion.

**Impact of weights.** Our choice of weights $\alpha_{ij}$ favors direct translation over indirect translation. In this set of experiments, we look at the impact of this choice over simple uniform weights. We compare UMH with uniform weights on direct and indirect word translation with a NN criterion. It is not surprising to see that uniform weights improve the quality of indirect word translation at the cost of poorer direct translation. In general, we experimentally found that our weighting makes UMH more robust when scaling to larger number of languages.

**Impact of the initialization.** In Sec. 2.2, we introduce a novel initialization based on the Gromov-Wassertein approach. Instead, Grave et al. (2018) consider a convex relaxation of Eq. (2) applied to centered vectors. Table 5 shows the impact of our initialization on the performance of UMH for direct and indirect bilingual alignment. We restrict this comparison to the 6 languages with existing

|                      | Direct   | Indirect |
|----------------------|----------|----------|
| Convex relaxation    | 77.8     | **71.5** |
| Gromov-Wasserstein   | **78.6** | 69.8     |

Table 5: Comparison of two different initializations for UMH on direct and indirect translations with a NN criterion. Convex relaxation refers to the initialization of Grave et al. (2018), while ours is Gromov-Wasserstein. We consider the 6 languages with mutual MUSE bilingual lexicons. We only learn bilingual mappings to and from English and translate with English as a pivot.

mutual bilingual lexicons in MUSE, i.e., English, French, Italian, Portuguese, Spanish and German. We consider only direct translation to and from English and the rest of the language pairs as indirect translation. Our initialization (Gromov-Wasserstein) outperforms the convex relaxation on what it is optimized for but this leads to a drop of performance on indirect translation. The most probable explanation for this difference of performance is the centering of the vectors.

|          | # languages | time | en       | de       | fr       | es       | it       | pt       | Avg.     |
|----------|-------------|------|----------|----------|----------|----------|----------|----------|----------|
| MAT+MPSR | 6           | 5h   | 79.6     | **70.5** | **82.0** | **82.9** | **80.9** | **80.1** | **79.3** |
| UMH      | 6           | 2h   | **80.7** | 69.1     | 81.0     | 82.2     | 79.8     | 78.9     | 78.6     |
| UMH      | 11          | 5h   | 80.4     | 68.3     | 80.5     | 81.7     | 79.0     | 78.2     | 78.0     |

Table 6: Impact of the number of languages on UMH performance. We report accuracy with a CSLS criterion on the 6 common languages. We average accuracy of translations from and to a single language. Numbers for MAT+MPSR are from Chen and Cardie (2018).

**Impact of the number of languages.** Table 6 shows the impact of the number of languages on UMH. We train our models on 6 and 11 languages, and test them on the 6 common languages as in Chen and Cardie (2018). We use the same default hyper-parameters for all our experiments. These 6 languages are English, German, French, Spanish, Italian and Portuguese. They are relatively simple to align together. Adding new and distant languages only affects the performance of UMH by less than a percent. This shows that UMH is robust to an increasing number of languages, even when these additional languages are quite distant from the 6 original ones. Finally, MAT+MPSR is slightly better than UMH ($+0.7\%$) on indirect translation. This difference is caused by our non uniform weights $\alpha_{ij}$ that seems to have a stronger impact on small number of languages. Note that our approach is computationally more efficient, training in 2h on a CPU instead of 5h on a GPU.

**Language tree.** We compute a minimum spanning tree on the matrix of losses given by UMH between every pairs of languages, except English. Three clusters appear: the Latin, Germanic and Slavic families chained as Latin-Germanic-Slavic. This qualitiative result is coherent with Table 3. However, the edges between languages make little sense, e.g., the edge between Spanish and Dutch. Our alignment based solely on embeddings is too coarse to learn subtle relations between languages.

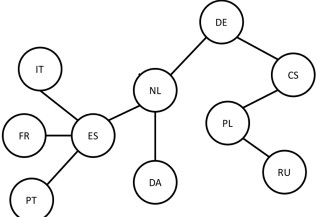

## 6 CONCLUSION

This paper introduces an unsupervised multilingual alignment method that maps every language into a common space while minimizing the impact on indirect word translation. We show that a simple extension of a bilingual formulation significantly reduces the drop of performance of indirect word translation. Our multilingual approach also matches the performance of previously published bilingual and multilingual approaches on direct translation. However, our current approach scales relatively well with the number of languages, but it is not clear if such a simple approach would be enough to jointly learn the alignment of hundreds of languages.

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

## APPENDIX

We present detailed results of our experiments on 6 and 11 languages.

| | en-fr | en-es | en-it | en-pt | en-de | en-da | en-nl | en-pl | en-ru | en-cs |
|---|---|---|---|---|---|---|---|---|---|---|
| *unsupervised, unconstrained* | | | | | | | | | | |
| Bil. - NN | 80.1 | 80.9 | 76.5 | 77.7 | 73.1 | 55.7 | 72.9 | 55.9 | 43.7 | 49.3 |
| Bil. - CSLS | 82.9 | 82.5 | 79.4 | 81.7 | 74.8 | 61.7 | 76.7 | 57.7 | 45.3 | 52.6 |
| *unsupervised, constrained* | | | | | | | | | | |
| Mul. - NN | 79.7 | 81.3 | 76.2 | 78.1 | 73.3 | 55.8 | 71.6 | 54.2 | 44.1 | 50.9 |
| Mul. - CSLS | 82.7 | 82.4 | 78.1 | 81.3 | 75.5 | 60.4 | 76.3 | 56.1 | 45.8 | 53.5 |
| | fr-en | es-en | it-en | pt-en | de-en | pl-en | ru-en | da-en | nl-en | cs-en |
| *unsupervised, unconstrained* | | | | | | | | | | |
| Bil. - NN | 80.3 | 82.5 | 77.9 | 79.7 | 71.4 | 63.9 | 74.2 | 67.7 | 60.7 | 61.3 |
| Bil. - CSLS | 83.3 | 84.9 | 79.4 | 82.2 | 73.7 | 65.9 | 77.1 | 67.7 | 62.8 | 62.5 |
| *unsupervised, constrained* | | | | | | | | | | |
| Mul. - NN | 80.6 | 81.8 | 77.3 | 78.8 | 72.1 | 64.6 | 73.1 | 68.3 | 62.3 | 63.9 |
| Mul. - CSLS | 83.4 | 85.1 | 79.3 | 81.4 | 74.4 | 67.0 | 76.2 | 68.9 | 64.9 | 64.4 |

Table 7: Full results on direct translation with the 11 languages and both NN and CSLS criteria for the UMH method. Bil. stands for bilingual, and Mul. stands for multilingual.

| → | fr | es | it | pt | de | da | nl | pl | ru | cs |
|---|---|---|---|---|---|---|---|---|---|---|
| fr | - | 82.5 | 82.1 | 77.1 | 69.3 | 53.1 | 67.8 | 47.1 | 40.7 | 42.7 |
| es | 84.9 | - | 83.3 | 86.5 | 68.3 | 54.7 | 66.6 | 48.6 | 44.2 | 46.7 |
| it | 86.5 | 86.3 | - | 79.8 | 66.6 | 51.8 | 66.0 | 50.8 | 39.4 | 43.9 |
| pt | 82.9 | 91.5 | 79.9 | - | 63.0 | 52.2 | 63.7 | 48.8 | 39.7 | 44.4 |
| de | 73.0 | 66.4 | 68.5 | 58.6 | - | 59.5 | 69.8 | 48.6 | 39.8 | 44.9 |
| da | 59.1 | 63.4 | 59.1 | 61.0 | 65.4 | - | 65.4 | 44.5 | 34.0 | 42.2 |
| nl | 69.0 | 70.1 | 68.6 | 67.7 | 75.9 | 58.8 | - | 47.7 | 40.4 | 44.9 |
| pl | 62.5 | 66.1 | 61.4 | 63.3 | 62.3 | 49.3 | 59.9 | - | 53.3 | 57.7 |
| ru | 60.1 | 61.4 | 57.1 | 57.3 | 55.9 | 46.0 | 54.2 | 56.3 | - | 52.2 |
| cs | 60.7 | 62.9 | 59.4 | 61.4 | 59.6 | 53.6 | 58.5 | 59.7 | 49.2 | - |

Table 8: Full results on indirect translation with the 11 languages with a CSLS criterion.

| → | en | fr | es | it | pt | de |
|---|---|---|---|---|---|---|
| en | - | 82.7 | 82.5 | 78.9 | 82.0 | 75.1 |
| fr | 83.1 | - | 82.7 | 82.5 | 77.5 | 69.8 |
| es | 85.3 | 85.1 | - | 83.3 | 86.3 | 68.7 |
| it | 79.9 | 86.7 | 87.0 | - | 80.4 | 67.5 |
| pt | 82.1 | 83.6 | 91.7 | 81.1 | - | 64.4 |
| de | 75.5 | 73.5 | 67.2 | 68.7 | 59.0 | - |

Table 9: Full results of our model trained on 6 languages with a CSLS criterion.

