# OpenReview forum: "Unsupervised Hyper-alignment for Multilingual Word Embeddings"
_ICLR.cc/2019/Conference_

### Official Review · AnonReviewer1 · 2018-10-30
**Good work for multilingual embedding alignment**

**Rating:** 7
**Confidence:** 3

**Review:**

This is a work regarding the alignment of word embedding for multiple languages.Though there are existing works similar to this one, most of them are only considering a pair of two languages, resulting in the composition issue mentioned in this work. The authors proposed a way of using a regularization term to reduce such degraded accuracy and demonstrate the validity of the proposed algorithm via experiments. I find the work to be interesting and well written. Several points that I want to bring up:

1. The language tree at the end of section 5 is very interesting. Does it change if the initialization/parameter is different?

2. The matrix P in (1) is simply a standard permutation matrix. I think the definitions are redundant.

3. The experiment results are expected since the algorithms are designed for better composition quality. An additional experiment, e.g. classification of instances in multiple languages, could further help demonstrate the strength of the proposed technic.

4. How to choose the regularization parameter \mu and what's the effect of \mu?

5. Some written issues like the notation of orthogonal matrix set, both \mathcal{O} and \mathbb{O} are used.

---

### Official Review · AnonReviewer2 · 2018-10-31
**An interesting paper that needs more work, more examples and more motivation.**

**Rating:** 6
**Confidence:** 4

**Review:**

This paper is concerned with the idea of inducing multilingual word embeddings (i.e., word vector spaces where words from more than two languages are represented) in an unsupervised way using a mapping-based approach. The main novelty of the work is a method, inspired by recent work of Nakashole and Flauger, and building on the unsupervised bilingual framework of Grave et al., which aims at bypassing the straightforward idea of independently mapping N-1 vector spaces to the N-th pivot space by adding constraints to ensure that the learned mappings can be composed (btw., it is not clear from the abstract what this means exactly).

In summary, this is an interesting paper, but my impression is that it needs more work to distinguish itself from prior work and stress the contribution more clearly.

Although 11 languages are used in evaluation, the authors still limit the evaluation only to (arguably) very similar languages (all languages are Indo-European and there are no outliers, distant languages or languages from other families at all, not even the usual suspects like Finnish and Hungarian). Given the observed instability of GAN-based unsupervised bilingual embedding learning, dissected in Sogaard et al.'s paper (ACL 2018) and also touched upon in the work of Artetxe et al. (ACL 2018), one of the critical questions for this work should also be: is the proposed method stable? What are the (in)stability criteria? When does the method fail and can it lead to sub-optimal solutions? What is the decrease in performance when moving to a more distant language like Finnish, Hungarian, or Turkish? Is the method more robust than GAN-based models? All this has to be at least discussed in the paper.

Another question is: do we really want to go 'fully unsupervised' given that even a light and cheap source of supervision (e.g., shared numerals, cognates) can already result in more robust solutions? See the work of Artetxe et al. (ACL 2017, ACL 2018), Vulic and Korhonen (ACL 2016) or Sogaard et al. (ACL 2018) for some analyses on how the amount of bilingual supervision can yield more (or less) robust models? Is the proposed framework also applicable in weakly-supervised settings? Can such settings with weak supervision guarantee increased robustness (and maybe even better performance)? I have to be convinced more strongly: why do we need fully unsupervised multilingual models, especially when evaluation is conducted only with resource-rich languages?

Another straightforward question is: can the proposed framework handle cases where there exists supervision for some language pairs while other pairs lack supervision? How would the proposed framework adapt to such scenarios? This might be an interesting point to discuss further in Section 5.

Style and terminology: it is not immediately clear what is meant by (triplet) constraints (which is one of the central terms in the whole work). It is also not immediately clear what is meant by composed mappings, hyper-alignment (before Section 4), etc. There is also some confusion regarding the term alignment as it can define mappings between monolingual word embedding spaces as well as word-level links/alignments. Perhaps, using mapping instead of alignment might make the description more clear. In either case, I suggest to clearly define the key concepts for the paper. Also, the paper would contribute immensely from some running examples illustrating the main ideas (and maybe an illustrative figure similar to the ones presented in, e.g., Conneau et al.'s work or Lample et al.'s work). The paper concerns word translation and cross-lingual word embeddings, and there isn't a single example that serves to clarify the main intuition and lead the reader through the paper. The paper is perhaps too much focused on the technical execution of the idea to my own liking, forgetting to motivate the bigger picture.

Other: the part on "Language tree" prior to "Conclusion" is not useful at all and does not contribute to the overall discussion. This could be safely removed and the space in the paper should be used to additional comparisons with more baselines (see above for some baselines).

The authors mention that their approach is "relatively hard to scale" only in their conclusion, while algorithmic complexity remains one of the key questions related to this work. I would like to see some quantitative (time) measurements related to the scaling problem, and a more thorough explanation why the method is hard to scale. The complexity and non-scalability of the method was one of my main concerns while reading the paper and I am puzzled to see some remarks on this aspect only at the very end of the paper. Going back to algorithmic complexity, I think that this is a very important aspect of the method to discuss explicitly. The authors should provide, e.g., O-notation complexity for the three variant models from Figure 2 and help the reader understand pros and cons of each design also when it comes to their design complexity. Is the only reason to move from the star model to the HUG model computational complexity? This argument has to be stressed more strongly in the paper.

Two very relevant papers have not be cited nor compared against. The work of Artetxe et al. (ACL 2018) is an unsupervised bilingual word embedding model similar to the MUSE model of Conneau et al. (ICLR 2018) which seems more robust when applied on distant languages. Again, going back to my previous comment, I would like to see how well HUG fares in such more challenging settings. Further, a recent work of Chen and Cardie (EMNLP 2018) is a multilingual extension of the bilingual GAN-based model of Conneau et al. Given that the main goal of this work and Chen and Cardie's work is the same: obtaining multilingual word embeddings, I wonder how the two approHowaches compare to each other. Another, more general comment concerns the actual evaluation task: as prior work, it seems that the authors optimise and evaluate their embeddings solely on the (intrinsic) word translation task, but if the main goal of this research is to boost downstream tasks in low-resource languages, I would expect additional evaluation tasks beyond word translation to make the paper more complete and convincing.

The method relies on a wide spectrum of hyper-parameters. How are these hyper-parameters set? How sensitive is the method to different hparams configurations? For instance, why is the Gromov-Wasserstein approach applied only to the first 2k vectors? How are the learning rate and the batch size determined?

Minor:
What is W in line 5 of Algorithm 1?
Given the large number of symbols used in the paper, maybe a table of symbols put somewhere at the beginning of the paper would make the paper easier and more pleasant to read.
I would also compare the work to another relevant supervised baseline: the work from Smith et al. (ICLR 2017). This comparison might further strengthen the main claim of the paper that indirect translations can also be found without degrading performance in multilingual embedding spaces.

---

### Official Review · AnonReviewer3 · 2018-11-04
**Need some more clarifications on the experiments**

**Rating:** 5
**Confidence:** 3

**Review:**

The authors present a method for unsupervised alignment of word across multiple languages. In particular, they extend an existing unsupervised bilingual alignment to the case of multiple languages by adding constraints to the optimization problem. The main aim is to ensure that the embeddings can now be composed and the performance (alignment quality) does not degrade across multiple compositions.

Strengths
- Very clearly written
- A nice overview of existing methods and correct positioning of the author's contributions in the context of these works
- A good experimental setup involving multiple languages

Weaknesses
- I am not sure how to interpret the results in Table 2 and Table 3 (see questions below).

Questions
- On page 7 you have mentioned that "this setting is unfair to the MST baseline, since ...." Can you please elaborate on this? I am not sure I understand this correctly.

- Regarding results in Table 2 and 3: It seems that there is a trade-off while adding constraints which results in poor bilingual translation quality. I am not sure is this is acceptable. I understand that your goal is to do indirect translation but does that mean we should ignore direct translation ?

- In Table 3 can you report both W-Proc and W-Proc* results ? Is it possible that the GW-initialization helps bilingual translation as the performance of W-Proc* is clearly better than W-Proc in Table 2. However, could it be the case that this somehow affects the performance in the indirect translation case? IMO, this is worth confirming.

- In Table 3, you are reporting  average accuracies across and within families. I would like to see the numbers for all language pairs independently. This is important because when you consider the average it is quite likely that for some language pair the numbers were much higher which tilts the average in favor of some approach. Also looking at the individual numbers will help us get some insights into the behavior across language pairs.

- In the motivation (Figure 1) it was mentioned that compositions can be done (and are often desirable) along longer paths (En-Fr-Ru-It). However, in the final experiments the composition is only along a triplet (X-En-Y). Is that correct or did I misinterpret the results? If so, can you report the results when the number of compositions increases?

---

### Comment · Area_Chair1 · 2018-11-19
**Reviewers: Please take a look at the author response**

Dear reviewers: could you please take a look at the author response? I think it is comprehensive, and could very well address some of the concerns expressed in the original reviews. I'd appreciate any additional feedback or discussion, which would help write the final review of the paper.

---

### Meta-Review · Area_Chair1 · 2018-12-13
**Simple and effective method, accuracy worse but speed better than contemporaneous work.**

**Confidence:** 2
**Recommendation:** Accept (Poster)

**Metareview:**

This paper provides a simple and intuitive method for learning multilingual word embeddings that makes it possible to softly encourage the model to align the spaces of non-English language pairs. The results are better than learning just pairwise embeddings with English.

The main remaining concern (in my mind) after the author response is that the method is less accurate empirically than Chen and Cardie (2018). I think however that given that these two works are largely contemporaneous, the methods are appreciably different, and the proposed method also has advantages with respect to speed, that the paper here is still a reasonably candidate for acceptance at ICLR.

However, I would like to request that in the final version the authors feature Chen and Cardie (2018) more prominently in the introduction and discuss the theoretical and empirical differences between the two methods. This will make sure that readers get the full picture of the two works and understand their relative differences and advantages/disadvantages.